# Foot and Mouth Disease Vaccine Matching and Post-Vaccination Assessment in Abu Dhabi, United Arab Emirates

**DOI:** 10.3390/vetsci11060272

**Published:** 2024-06-14

**Authors:** Yassir M. Eltahir, Hassan Zackaria Ali Ishag, Krupali Parekh, Britta A. Wood, Anna Ludi, Donald P. King, Oum Keltoum Bensalah, Rashid A. Khan, Asma Abdi Mohamed Shah, Kaltham Kayaf, Meera Saeed Mohamed

**Affiliations:** 1Animals Extension and Health Services Division, Abu Dhabi Agriculture and Food Safety Authority (ADAFSA), Abu Dhabi P.O. Box 52150, United Arab Emirates; 2Biosecurity Affairs Division, Development and Innovation Sector, Abu Dhabi Agriculture and Food Safety Authority (ADAFSA), Abu Dhabi P.O. Box 52150, United Arab Emirates; 3FAO World Reference Laboratory for FMD (WRLFMD), The Pirbright Institute, Pirbright GU24 0NF, UK; 4Animal Development and Health Department, Ministry of Climate Change and Environment, Dubai P.O. Box 1509, United Arab Emirates

**Keywords:** foot and mouth disease virus, United Arab Emirates, vaccine matching, post-vaccination assessment

## Abstract

**Simple Summary:**

Livestock in the United Arab Emirates (UAE) undergo annual vaccination against foot and mouth disease (FMD). The UAE animal health plan centers on the use of FMD vaccines to minimize disease impacts and control the spread of the disease. In this study, serotype O FMD virus (FMDV) isolates collected from outbreaks in 2021 were subjected to a vaccine matching analysis against six serotype O vaccine strains. Additionally, post-vaccination coverage for serotypes A and O of FMDV was evaluated using a solid-phase competitive ELISA. The findings indicate that the FMD vaccinal strains utilized in the Abu Dhabi Emirate were antigenically matched with the field isolates. Moreover, the implemented FMD vaccination program with a booster dose elicited FMDV-specific antibody responses in sheep and goat herds with >80% coverage.

**Abstract:**

Despite the annual vaccination of livestock against foot and mouth disease (FMD) in the United Arab Emirates (UAE), outbreaks of the disease continue to be reported. The effective control of field outbreaks by vaccination requires that the vaccines used are antigenically matched to circulating field FMD viruses. In this study, a vaccine matching analysis was performed using the two-dimensional virus neutralization test (VNT) for three field isolates belonging to the O/ME-SA/PanAsia-2/ANT-10 and O/ME-SA/SA-2018 lineages collected from different FMD outbreaks that occurred within the Abu Dhabi Emirate in 2021 affecting *Arabian oryx* (*Oryx leucoryx*), goat, and sheep. In addition, post-vaccination antibodies in sheep and goats were measured using solid-phase competitive ELISA (SPCE) for FMDV serotypes A and O at five months after a single vaccine dose and a further 28 days later after a second dose of the FMD vaccine. An analysis of vaccine matching revealed that five out of the six vaccine strains tested were antigenically matched to the UAE field isolates, with *r*_1_-values ranging between 0.32 and 0.75. These results suggest that the vaccine strains (O-3039 and O1 Manisa) included in the FMD vaccine used in the Abu Dhabi Emirate are likely to provide protection against outbreaks caused by the circulating O/ME-SA/PanAsia-2/ANT-10 and O/ME-SA/SA-2018 lineages. All critical residues at site 1 and site 3 of VP1 were conserved in all isolates, although an analysis of the VP1-encoding sequences revealed 14–16 amino acid substitutions compared to the sequence of the O1 Manisa vaccine strain. This study also reports on the results of post-vaccination monitoring where the immunization coverage rates against FMDV serotypes A and O were 47% and 69% five months after the first dose of the FMD vaccine, and they were increased to 81 and 88%, respectively, 28 days after the second dose of the vaccine. These results reinforce the importance of using a second booster dose to maximize the impact of vaccination. In conclusion, the vaccine strains currently used in Abu Dhabi are antigenically matched to circulating field isolates from two serotype O clades (O/ME-SA/PanAsia-2/ANT-10 sublineage and O/ME-SA/SA-2018 lineage). The bi-annual vaccination schedule for FMD in the Abu Dhabi Emirate has the potential to establish a sufficient herd immunity, especially when complemented by additional biosecurity measures for comprehensive FMD control. These findings are pivotal for the successful implementation of the region’s vaccination-based FMD control policy, showing that high vaccination coverage and the wide-spread use of booster doses in susceptible herds is required to achieve a high level of FMDV-specific antibodies in vaccinated animals.

## 1. Introduction

Foot and mouth disease (FMD) is caused by a virus (FMDV) belonging to the genus Aphthovirus in the family Picornaviridae. It is a highly contagious disease that affects all cloven-hoofed animals. FMDV is transmitted through direct contact between animals, animal products (such as milk, meat, and semen), mechanical transfer via people or fomites, and via the airborne route [1,2]. The incubation period following infection ranges from 2 to 21 days (average 3–8 days) depending on factors such the species of the animal, infectious dose, serotype, and strain of the virus [3]. Affected animals commonly exhibit symptoms such as fever; the cessation of rumination; excessive salivation; and the appearance of blisters on the lips, tongue, mouth, nose, between the toes, and occasionally on the teats. Additionally, there may be a decrease in milk production. Young animals infected with FMD may experience higher mortality rates due to myocarditis, while those that recover from the disease may become carriers [3]. Although the fatality rate associated with FMD infection is low, it significantly impacts the economic capability of countries where the disease is endemic by reducing productivity and hindering the export of livestock and livestock products [4,5]. FMDV is classified into seven immunologically distinct serotypes (O, A, C, Asia 1, SAT 1, SAT 2, and SAT 3) distributed across seven geographic virus pools (1–7) [6,7]. The United Arab Emirates (UAE), located on the Arabian Peninsula within Pool 3, hosts FMDV serotypes O, A, and Asia 1. The predominant topotypes/lineages currently prevalent in this region include O/ME-SA/PanAsia-2, A/ASIA/Iran-05, and Asia 1/Sindh-08 [8,9]. However, the region has also encountered introductions of O/ME-SA/Ind-2001, O/ME-SA/SA-2018, and A/ASIA/G-VII from Pool 2 (South Asia), and SAT1/I and SAT 2/XIV from Pool 4 (East Africa) [10,11,12,13,14]. The current animal population in the UAE is estimated to be around 5 million animals, comprising 4.35 million small ruminants, 0.5 million camels, and 111,000 cattle. FMD is endemic in the UAE, affecting both domestic livestock and wildlife. It is classified as a notifiable disease in the UAE and was officially reported to the World Animal Health Information System (WAHIS) for the first time in 2003. To date, a total of 30 FMD outbreaks have been reported to WAHIS [15]. Serotype O is considered predominant in the UAE, and recent FMD outbreaks reported to WAHIS in 2021 have been caused by two different lineages of FMDV, namely O/ME-SA/SA-2018 and O/ME-SA/PanAsia-2 [16].

In regions where FMD is endemic or where outbreaks are highly likely, prophylactic vaccination is commonly employed [17]. FMD exhibits frequent spontaneous mutations, leading to considerable antigenic variability and the emergence of new topotypes and lineages, which can occasionally result in vaccination failure [18]. This antigenic diversity both among and within serotypes hampers the cross-reactivity of immune responses elicited by one FMDV strain against another, thereby limiting potential cross-protection. Thus, assessing the antigenic and immunogenic similarities between the vaccine strain and circulating field strain and ensuring their matching are essential for optimizing vaccination programs [19]. The expected level of protection provided by a vaccine is often measured by in vitro vaccine matching testing, which compares the seroreactivity of vaccine antisera to the vaccine strain (homologous reactivity) and the field strains (heterologous reactivity) [20].

The monitoring and evaluation of national FMD control strategies is a key component of the progressive control pathway for FMD (PCP-FMD) of the Global FMD Control and Eradication Strategy and the regional Middle East FMD control roadmap adopted by the FAO and the World Organization for Animal Health (WOAH). The UAE is at stage 2 (out of 5 stages) of the PCP-FMD, which implies the implementation of a risk-based control plan [2,3]. This requires continual monitoring of outbreak strains, the evaluation of FMD risks, the assessment of implementation levels, and control methods. Principles and guidelines for advising countries on Post-Vaccination Monitoring (PVM) procedures are published by the FAO and the WOAH [21].

The national animal health plan of the UAE aims to control and eradicate FMD from the country by 2030, aligning with the PCP-FMD. The plan involves the mass vaccination of cattle and small ruminants against FMD. Despite bi-annual livestock vaccination efforts, FMD cases continue to occur in the UAE, raising concern about the introduction of new viral strains, the lack of antigenic matching of the FMD vaccine used, and the efficacy of livestock vaccination campaigns. Therefore, the objectives of this study were to evaluate whether FMDV field strains causing outbreaks in the UAE are antigenically matched to the commercial FMD vaccine used and to assess the herd immunity induced in small ruminants by a single dose or two doses of the FMD vaccination regime commonly practiced in the Abu Dhabi Emirate. Three FMDV field isolates collected from three different outbreaks within Abu Dhabi Emirate in 2021 were tested against strains of the vaccine used, and their VP1 sequences were further characterized and analyzed. Moreover, a small-scale trial was conducted to assess FMD vaccination coverage in sheep and goats vaccinated with a commercial FMD vaccine in 2023. 

## 2. Materials and Methods

### 2.1. Field Isolates Used for Vaccine Matching

In 2021, three suspected outbreaks of FMD in the Abu Dhabi and Al Ain regions of the Abu Dhabi Emirate were reported in two (A and B) non-FMD-vaccinated farms and one (C) farm which received a single dose of FMD vaccination [16]. These farms kept various animal species, and after outbreak investigations, samples were collected including two mouth swabs (in a viral transport media) from clinically affected animals exhibiting symptoms such as fever, lameness, and vesicular lesion tissue (from an *Arabian oryx* (*Oryx leucoryx*) and one goat), and one heart tissue from a sheep (in a plain container) (Table 1). The specimens were placed on an ice box and immediately transported to the Abu Dhabi Agriculture and Food Safety Authority (ADAFSA) veterinary laboratories for molecular diagnosis of FMD. At ADAFSA, the samples underwent testing using RT-qPCR following the methodology described previously [16]. All samples were tested for the presence of FMDV where a RT-qPCR threshold of >39 indicated a positive sample. 

Three serotype O isolates recovered from the World Reference Laboratory-Line Fetal Bovine Kidney (WRL-LFBK) cells [22,23] were used in this study for vaccine matching at the FAO WRL for FMD (WRLFMD, Pirbright, United Kingdom) (Table 1). The VP1 sequences for these isolates were generated as previously described [16] and were deposited in the GenBank.

### 2.2. Analysis of Amino Acid Sequence Variability

The amino acid sequences of the FMD isolates were translated using Geneious Prime version 2023.1. To compare the amino acid sequences for field isolates with the O_1_ Manisa vaccine strain (GenBank: AY593823) used in the Abu Dhabi Emirate, individual multiple sequence alignment was prepared using the Geneious alignment tool within the Geneious software. It should be noted that the sequence and identity of the other serotype O vaccine strain used in the UAE (O-3039) has not been publicly disclosed. The variability in amino acid sequences, particularly focusing on the critical residues at the BC loops, GH loops, and C-termini of VP1 at the sites 1 and 3 in the VP1 coding sequences of the field strains, were evaluated in comparison to the sequence of the O_1_ Manisa vaccine strain.

### 2.3. Two-Dimensional Virus Neutralization Assay (2D-VNT) for Vaccine Matching

The serum utilized for the vaccine matching assay comprised a pool of sera from five vaccinated cattle, administered with a monovalent vaccine. This serum was collected 21 days post-vaccination (except O1 Campos, Biogénesis Bagó, which was collected 30 days post-vaccination), and subsequently tested against both the homologous (six different serotype O commercial vaccinal strains produced by three different companies) and heterologous (field) virus.[24]. Neutralization titers were determined from the regression data, representing the log_10_ reciprocal antibody dilution required for 50% neutralization of 100 tissue culture infective units of virus (log_10_SN50/100 Tissue Culture Infectious Dose 50 (TCID^50^). The antigenic relationship between the vaccine virus and the field virus was calculated using the ‘*r*_1_’ ratio: neutralizing the antibody titer against the field virus divided by the neutralizing antibody titer against the homologous virus. An *r*_1_-value greater than 0.3 indicated a close antigenic relationship between the field isolate and the vaccine strain, suggesting the likelihood of effective protection conferred by a potent vaccine containing the vaccine strain [24].

### 2.4. Post-Vaccination Assessment

#### 2.4.1. Study Area

In April–June 2023, sero-monitoring post-vaccination was applied to estimate vaccination coverage and the level of FMDV-specific antibodies on sheep and goat farms after a single dose or two doses of FMD vaccination at the Al Alain region of the Abu Dhabi Emirate where the last FMD outbreaks were reported in 2021. The region harbors 1,720,164 sheep and goats distributed on 13,159 holdings, representing 65% of the total ruminant population within the Abu Dhabi Emirate.

#### 2.4.2. Vaccine and Vaccination

Since 2019, a saponin-adjuvanted, NSP-purified, inactivated hexavalent FMD vaccine produced by Boehringer Ingelheim Animal Health (Pirbright, UK) and supplied by International Free Trade (IFT, Dubai, UAE) has been used to vaccinate livestock against FMD in the Abu Dhabi Emirate. This vaccine contains the following strains with a potency of over 6 PD50 per dose: O1 Manisa, O-3039, A Iran 05, A-GVII, Asia 1 Shamir, and SAT2 Eritrea. The vaccination regime comprises two doses (1 mL 4–6 months interval) for sheep and goats and three doses (3 mL 4 months interval) for cattle. The target of the FMD vaccination campaign coverage in 2023 was 85%. 

#### 2.4.3. Sample Size and Collection

The sizes of the animals’ holdings for sample collection were estimated using a two-stage cluster sampling method, aiming for a 95% confidence level, 5% precision, and an expected sero-prevalence of 80% [25]. Sixty-six individual mixed holdings of sheep and goat were selected using simple random sampling. In total, 396 serum samples were collected from adult (above 3 months of age) sheep and goats between April and May 2023. Six samples were collected from each small ruminant farm. Samples were collected at five months and 28 days after the first and the second FMD vaccination doses, respectively.The selected animals were individually identified to ensure accurate monitoring and were observed for clinical signs of FMD. Blood samples were obtained by venipuncture from the jugular vein. The animal blood samples were drawn into serum separator vacutainers with a red cap and transported to the ADAFSA laboratory in cold boxes within 24 h. Upon reaching the laboratory, the serum tubes were centrifuged at 3500 rpm for 5 min at room temperature, and the sera was divided into aliquots and then stored at 4 °C until testing.

#### 2.4.4. Serological Testing

Serum samples underwent testing for structural antibodies to FMDV serotypes A and O using commercially available solid-phase competitive ELISA (SPCE) following the guidelines provided by the manufacturer (Istituto Zooprofilattico Sperimentale della Lombardia e dell’Emilia Romagna (IZSLER), Brescia, Italy).

## 3. Results

### 3.1. Phylogenetic Analysis

The VP1 sequence of the FMD viruses obtained from *Arabian oryx* (UAE/1/2021) belonged to the O/ME-SA/PanAsia-2/ANT-10 sublineage and shared 100% nucleotide identity to isolate UAE/2/2021, which has previously been described [16]. FMDV isolates from goats (UAE/9/2021) and sheep (UAE/15/2021) were identified to belong to the O/ME-SA/SA-2018 lineage and shared 100% nucleotide identities to UAE/10/2021 and UAE/14/2021, respectively, which has previously been reported [16]. 

### 3.2. Amino Acid Sequence Variability Analysis for Identified Viruses

The sequence for isolate UAE/1/2021 (OR425051), exhibited amino acid sequence variability across 16 positions when compared to the O1 Manisa vaccine strain (GenBank: AY593823). Notably, 10 out of 16 amino acid substitutions were located at antigenic site 1 (GH loop 138–156), as outlined in Table 2. 

The other two isolates (UAE/9/2021 and OR425053) and UAE/15/2021 and OR425057 also demonstrated comparable amino acid sequence variability compared to the O1 Manisa vaccine strain, encompassing substitutions at 16 and 14 positions, respectively. Of these amino acid substitutions, 8/16 (50%) and 7/14 (50%) were located at site 1 of UAE/9/2021 and UAE/15/2021, respectively, as indicated in Table 2 and Figure 1. As expected, the tripeptide sequence Arg-Gly-Asp (RGD), which forms the integrin binding cellular receptor for FMDV (located at position 146–148 of the VP1 amino acid sequence), was entirely conserved in all sequences. Furthermore, the critical residues of VP1 at site 1 [ßG-ßH loop (residues 144, 148 and 154) and carboxy terminus (residue 208)] and at site 3 (residues 43 and 44 of the ßB-ßC loop) were conserved in all isolates compared to the O1 Manisa vaccine strain (AY593823).

### 3.3. Vaccine Matching with 2 dm VNT

The results of the two-dimensional virus neutralization test revealed that the O1 Campos (Biogénesis Bagó), O-3039( Boehringer Ingelheim), O Manisa (Boehringer Ingelheim), O PanAsia-2 (Boehringer Ingelheim), and O/TUR/5/09 (MSD Animal Health) vaccine strains were antigenically matched to the O/ME-SA/PanAsia-2/ANT-10 sublineage and O/ME-SA/SA-2018 lineage isolates, with *r*_1_-values ranging between 0.32–0.48 and 0.32–0.75, respectively. However, poor vaccine matching results were observed for the O_1_ Campos (Boehringer Ingelheim) vaccine strain against both lineages (had *r*_1_-values < 0.3) [Table 3]. Specifically, the two FMD vaccinal strains (O-3039, Boehringer Ingelheim; O Manisa, Boehringer Ingelheim) included in the vaccine used by ADAFSA exhibited antigenic matching with all three field isolates, causing FMD outbreaks with *r*_1_-values ranging from 0.38 to 0.75.

### 3.4. Evaluation of FMD Vaccination in 2023

During blood sample collection, a clinical examination of the presence of FMD symptoms indicated an absence of infection in the targeted animal holdings. Across all species tested, the seroprevalence rates at five months after the first initial FMD vaccination dose against FMDV serotypes A and O were 47% and 69%, respectively (Table 4). This prevalence increased to 81 and 88%, respectively, 28 days after the second FMD vaccination dose. Specifically, for serotype A, the immunization coverage rates were 53% and 39% in sheep and goats five months after the first vaccination, increasing to 78% and 86% in sheep and goats, respectively, 28 days after the second vaccination. Similarly, for serotype O, the immunization coverage rates of species were 72% and 47% in sheep and goats five months after the first vaccination, which increased to 92% and 82% in sheep and goats, respectively, 28 days after the second vaccination. Except for serotype O in sheep, the percentages of immunization coverage for both serotypes were below 70% in both species five months after the initial vaccination. 

## 4. Discussion

Vaccination stands as the primary strategy for controlling FMD in endemic regions like the UAE. The combination of vaccination and stamping out has proven effective in reducing or eradicating FMD from Europe and large parts of South America. Nonetheless, the highly contagious nature of FMDV, the presence of various circulating serotypes and their associated topotypes, the possible presence of wildlife reservoirs, and the continual emergence of new strains that are poorly matched to existing vaccines pose significant challenges for endemic countries in effectively controlling and mitigating the disease burden on both the national and regional levels [7]. The Sub-Saharan Africa and the Middle East–South Asia regions remain endemically affected by various FMD serotypes circulating extensively across these regions. This situation has profound implications for livestock production and poverty alleviation, hindering access to international markets, restricting genetic improvement, and impeding diary production development [26].

In the Abu Dhabi Emirate, and according to the national livestock vaccination program, sheep and goats undergo bi-annual vaccination against FMD using commercial vaccines to control the disease in these animal populations. The selection of FMDV strains included in the vaccine used in implementing the UAE national FMD control and eradication plan aligns with the recommendations regularly provided by the WRL-FMD and the updated outcomes of the UAE national animal health plan. Specifically, the current vaccine manufactured by Boehringer Ingelheim and used in the Abu Dhabi Emirate comprises FMDV strains O1 Manisa, O-3039, A Iran 05, A/ASIA/G-VII 2015, Asia 1 Shamir, and SAT2 Eritrea. 

The occurrence of FMD outbreaks in the UAE in 2021 [16] highlighted the potential for the emergence and circulation of new FMD viral strains that may not match the currently administered vaccine. This raises questions regarding the effectiveness of FMD vaccination among the targeted population in the Abu Dhabi Emirate in preventing FMD outbreaks. Regular post-vaccination assessments and vaccine matching analyses are essential requirements of the national animal health plan aimed at eradicating FMD from the UAE. However, these activities have not been previously reported. While in vivo vaccination–challenge experiments are considered the gold standard for FMD vaccine matching, they have limitations in terms of animal welfare, biosafety, and cost-effectiveness. In practice, FMD vaccine selection relies heavily on in vitro serological vaccine matching tests, such as virus neutralization tests (VNTs) and a liquid-phase blocking ELISA (LPBE) [27]. Therefore, in this study, samples confirmed to be infected with FMD were from an Arabian oryx, goat, and sheep, each originating from three different FMD outbreaks. Their VP1 coding sequences were analyzed to assess critical amino acid variability, and their antigen matching with FMD vaccinal strains used in the UAE was evaluated. Furthermore, the study included an assessment of FMD immunization coverage in vaccinated sheep and goat farms in a specific region of the Abu Dhabi Emirate.

The UAE-FMD isolate from *Arabian oryx* reported in this study was classified within the O/ME-SA/PanAsia-2/ANT-10 sublineage, while isolates from the sheep and goat were assigned to the O/ME-SA/SA-2018 lineage previously reported in the Abu Dhabi Emirate in 2021 [16]. Our investigation revealed that five out of six different FMDV-O vaccine strains tested were antigenically matched with these FMDV-O field strains. This is not surprising, as serotype O displays less antigenic diversity compared to other serotypes like A and SAT2 [28]. Of the five effective vaccine strains, two (O1 Manisa, Boehringer Ingelheim and O-3039, Boehringer Ingelheim) are currently utilized in the Abu Dhabi Emirate for FMD control. Notably, vaccine strains that exhibit poor matching with field strains may provide suboptimal protection [7]. Therefore, our findings suggest that the FMD-O vaccine strains employed in the Abu Dhabi Emirate antigenically matched the viruses that have caused recent FMD outbreaks [29].

The VP1 coding sequences were also analyzed to assess critical amino acid variability with one of the FMD vaccinal strains used in the VNT. Although there were several variations in amino acids, all residues critical at antigenic sites 1 and site 3 for VP1 [30,31,32], such as 148, 149, 154, and 208, were fully conserved across all isolates. All three isolates retained the conserved Arg-Gly-Asp cell attachment site [33,34,35], which is consistent with previous findings [30,36,37]. The amino acid change at position 139 has been reported to impact the serum neutralization of O isolate variants [38,39,40]. However, the substitution observed for the three field isolates appeared to have no effect on virus neutralization in the vaccine matching test (Table 3). It is noteworthy that the sequence of the O-3039 vaccine strain also used in the UAE is not publicly available. Therefore, we were unable to characterize the amino acid variability of UAE isolates in comparison to this vaccine strain.

Several limitations associated with the use of inactivated FMD vaccines, which are commonly employed to combat FMD in endemic regions, have been reported. These include the necessity for high-containment facilities to culture virulent FMDV, short-duration immunity, limited cross protection among various strains and topotypes within the same serotype, the frequent emergence of new variants capable of evading vaccine-induced immunity, and the inability to eliminate virus carriers [7]. The investigation of two outbreaks reported here that affected *Arabian oryx* and unvaccinated goat in two different farms highlights the risk posed by the cohabitation of wildlife with sheep and goats in the same farm, unrestricted animal movement between farms, and factors related to vaccination coverage in the spread and control of FMD in the Abu Dhabi Emirate. Therefore, it is crucial to implement measures such as avoiding the mixing of wildlife with livestock, controlling animal movement, increasing vaccination coverage, enhancing farms’ biosecurity measures, and regularly conducting post-vaccination assessments and vaccine matching for FMD field isolates in the Abu Dhabi Emirate. These measures will contribute to improving the performance of the national FMD control plan and facilitate the achievement of its objective’s requirements. 

FMD immunization coverage in vaccinated sheep and goat farms in the Al Ain region of the Abu Dhabi Emirate, where outbreaks occurred in 2021, was also assessed. One of the field isolates tested in this study originated from a sheep farm that received a single dose of FMD vaccination in 2021 [16]. This raised the need to investigate the duration of protective immunity after one or two rounds of FMD vaccination in small ruminants. The post-vaccination immunity assessment against FMD serotypes A and O conducted here revealed overall immunity rates of 47% and 69% in both vaccinated sheep and goat herds five months after the first vaccination, respectively. This immunity increased to 81% and 86% for FMD serotypes A and O, respectively, four weeks after the booster second of the FMD vaccine. To ensure the effectiveness of FMD vaccines used to control the disease in specific countries, or a region, post-vaccination monitoring is required. 

Generally, most commercial vaccines recommend an initial vaccination course consisting of two doses, typically administered one month apart, followed by a booster dose every 4–6 months. In an experimental study, neutralizing titers against an O/ME-SA/PanAsia strain were assessed following either a two-dose primary course (1 mL) or a single double-dose (2 mL) vaccination in sheep with a 6 PD50 vaccine. The results show that titers did not significantly differ between the groups except at six months post-vaccination, where the single double-dose group exhibited significantly lower titers below the established protective cut-off [41]. 

FMD vaccines are typically available in standard (3 PD50) and higher-potency (6 PD50) formulations, determined by the number of 50% protective doses in each dose. High-potency vaccines have potential application for emergency reactive campaigns in FMD-free areas due to their rapid immunity onset. They address challenges posed by antigenic variations among virus strains of the same serotype, potentially protecting against clinical FMD even when there is a mismatch with the circulating field strain [42]. Presently, specific criteria for achieving a protective level of FMD immunity have not been established, but it is advised to maintain at least 80% immune animals within susceptible populations [21,43]. 

Indeed, the planned vaccination coverage in the Abu Dhabi Emirate was set at 85% in 2023. The findings obtained here indicate that a single dose of small ruminant FMD vaccination alone may not suffice to reach the necessary level of protective immunity against FMD serotypes A and O. It further requires the administration of a booster dose to achieve the desired level of immunity of above 80%. Moreover, the bi-annual vaccination schedule for small ruminants in the Abu Dhabi Emirate appears effective in establishing an acceptable herd immunity level, especially when combined with other biosecurity measures for FMD control in the region. Consequently, there is a need for further investigations into the post-vaccination assessment of other FMD serotypes (SAT2) currently circulating in the region using VNT, as well as a need to determine the duration of protective immunity following the second vaccination.

## 5. Conclusions

This is the first report on FMDV vaccine matching and the first post-vaccination assessment conducted in the Abu Dhabi Emirate. The results from the virus neutralization test indicate that all of the FMD field isolates examined in this study (belonging to the O/ME-SA/PanAsia-2/ANT-10 sublineage and O/ME-SA/SA-2018 lineage) were matched with the vaccinal strains (O-3039 and O1 Manisa, Boehringer Ingelheim) included in the FMD vaccine used in the Abu Dhabi Emirate. Furthermore, a post-vaccination assessment against FMD serotypes A and O indicated that a protective herd immunity exceeding 80% could be achieved with the current bi-annual vaccination regime. A high vaccination coverage of up to 100% in susceptible herds, including sheep, goats, and cattle, coupled with post-vaccination sero-surveillance to other FMD serotypes circulating in the Arabian Peninsula, is required to monitor antibody titers in vaccinated animals to control FMD in the region and achieve the PCP-FMD and the regional Middle East FMD control roadmap requirements.

## Figures and Tables

**Figure 1 vetsci-11-00272-f001:**
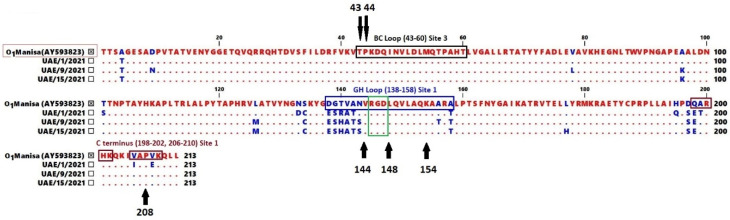
The amino acid sequence alignment reveals differences between the serotype O FMDV UAE field isolates and O1 Manisa, AY593823. The amino acids in dots are identical to the vaccine strain. The epitope regions of the virus, including site 1 [the GH-loop (138–158) in the blue box and C-terminus (198–202, 206–210) in the dark red boxes] as well as site 3 [the BH loop (positions at 43–60) in the black box] are shown. The motif RGD at positions 146–148 in site 1 is highlighted in the green box. Putative critical amino acids at each position are numbered and indicated by black arrows.

**Table 1 vetsci-11-00272-t001:** Isolates collected in UAE in 2021 for vaccine matching.

Farm Name	Date of Sample Collection	Sample Type	No. of Samples	WRLFMD Label	Animal Species Infected	FMD Vaccination	GenBank Accession Number
**A**	April 2021	Mouth swab	1	UAE/1/2021	*Arabian oryx*	*No*	OR425051
**B**	November 2021	Mouth swab	1	UAE/9/2021	Goat	No	OR425057
**C**	December 2021	Heart tissue	1	UAE/15/2021	Sheep	Yes-single dose *	OR425053

* The vaccine (Boehringer Ingelheim) used contained the following vaccine strains: O1 Manisa, O-3039, A Iran 05, A-GVII, Asia 1 Shamir, and SAT2 Eritrea.

**Table 2 vetsci-11-00272-t002:** VP1 amino acids’ variability per antigenic site of UAE isolates compared to O_1_ Manisa (AY593823) vaccine strain.

Isolate	Lineage (Sublineage)	Vaccine Strain Compared to	Number of Amino Acid Substitutions	Variability at Site 1			Variability at Site 3
				GH Loop	C-Terminus		BC Loop
				(138–156)	(198–202)	(206–210)	(43–60)
UAE/1/2021	O/ME-SA/PanAsia-2 (ANT-10)	O_1_ Manisa (AY593823)	16	6	2	2	-
UAE/9/2021	O/ME-SA/SA-2018		16	8	-	-	-
UAE/15/2021			14	7	-	-	-

**Table 3 vetsci-11-00272-t003:** The vaccine matching results obtained for UAE isolates UAE/1/2021, UAE/9/2021, and UAE/15/2021. For each field isolate, the *r*_1_-value is presented, followed by the heterologous neutralization titer (*r*_1_-value, titer). An *r*_1_ greater than 0.3 suggests a close antigenic relationship between the field isolate and the vaccine strain, indicating potential protection. Conversely, an *r*_1_-value less than 0.3 suggests an antigenic difference between the field isolate and the vaccine strain. The vaccine strains used in the Abu Dhabi Emirate are highlighted in gold.

Vaccine	O/UAE/1/2021 O/ME-SA/PanAsia-2/ANT-10	O/UAE/15/2021 O/ME-SA/SA-2018	O/UAE/9/2021 O/ME-SA/SA-2018
O_1_ Campos, Biogénesis Bagó	0.43, 2.56	0.51, 2.63	0.60, 2.70
O-3039, Boehringer Ingelheim	0.38, 1.64	0.75, 1.94	0.59, 1.83
O_1_ Campos, Boehringer Ingelheim	0.19, 2.02	0.28, 2.19	0.23, 2.10
O_1_ Manisa, Boehringer Ingelheim	0.48, 2.08	0.56, 2.15	0.44, 2.04
O PanAsia-2, Boehringer Ingelheim	0.32, 2.14	0.47, 2.30	0.32, 2.13
O/TUR/5/09, MSD Animal Health	0.44, 2.13	0.69, 2.32	0.68, 2.32

**Table 4 vetsci-11-00272-t004:** Evaluation of FMD vaccination in 2023 against serotypes A and O.

Period	Result	Serotype A		Serotype O		Serotype A in Total Sheep and Goats	Serotype O in Total Sheep and Goats
		Sheep	Goat	Sheep	Goat		
Five months after the 1st FMD vaccination dose	Total tested	230	166	230	166	396	396
	No. of positive	123	64	166	106	187	272
	No. of negative	107	102	64	60	209	124
% Of immunity coverage		53%	39%	72%	64%	47%	69%
	Total tested	230	166	230	166	396	396
28 days after the 2nd FMD vaccination dose	No. of positive	179	142	211	136	321	347
	No. of negative	51	24	19	30	75	105
% of immunity coverage		78%	86%	92%	82%	81%	88%

## Data Availability

The VP1 gene of the FMDV UAE strain sequences generated in this study are available in the NCBI database under the accession numbers mentioned in the manuscript.

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
