# Peer review of "Foot and Mouth Disease Vaccine Matching and Post-Vaccination Assessment in Abu Dhabi, United Arab Emirates"

_vetsci, 2024, doi:10.3390/vetsci11060272_

Round 1
Reviewer 1 Report
Comments and Suggestions for Authors
This manuscript is well-written. Results in FMDV vaccine matching and post-vaccination assessments conducted in their country are very clear. Results from the neutralization tests indicated that all the FMD isolates inthis study matched with the vaccinal strains included in the jn the vaccine used in their country. The post-vaccination assessements against FMD serotypes A & O indicated that a protective herd immunity exceeding 80% could be achieved with the current vaccination programme in their country.
However minor modifications in the manuscript are suggested.
1. L54 PVM could be Post Vaccination Monitoring (PVM).
2. L146 two sawbs could be two mouth swabs.
Author Response
Thanks for reviewing our manuscript and please find below our detailed response
Comments and Suggestions for Authors
However minor modifications in the manuscript are suggested.
- L54 PVM could be Post Vaccination Monitoring (PVM).
Changed and highlighted
- L146 two swabs could be two mouth swabs.
Added

Reviewer 2 Report
Comments and Suggestions for Authors
Foot‑and‑mouth disease vaccine matching and post-vaccination 2 assessment in the Emirate of Abu Dhabi, United Arab Emirates. The manuscript submitted by Eltahir et al. described „Foot‑and‑mouth disease vaccine matching and post-vaccination 2 assessment in the Emirate of Abu Dhabi, United Arab Emirates“. The manuscript is well-designed and well-written. I have only a few comments:
- The abstract is too long. No need for subtitles in the abstract (Conclusion)
Line 93: Please update the epidemiology with the most recent publication.
Abbreviations: Please expand the full names for all abbreviations that are mentioned for the first time.
Tables and figures should be stand-alone; please describe all abbreviations in the footnote or ligands.
Materials: Please provide the supplier, city, and Country of all materials used
Is serology enough for evaluating the efficacy of FMDV vaccines?
Additional minor comments: See attached pdf file

Minor editing of English language required
Author Response
Thanks for reviewing our manuscript and please find below our detailed response
I have only a few comments:
- The abstract is too long. No need for subtitles in the abstract (Conclusion)
Abstract follows the journal format. It was reduced , the subtitle was also removed and rearranged to be as follows:
Simple Summary: Livestock in the United Arab Emirates (UAE) undergo annual vaccination against Food and Mouth Disease (FMD). However, after implementing the UAE animal health plan which include FMD, matching of the FMD field isolates with the vaccines used and assessment of the post-vaccination efficacy have not done before. In this study, serotype O FMD virus (FMDV) isolates collected from outbreaks in 2021 were subjected to vaccine matching analysis against six serotype O vaccine strains. Additionally, a post-vaccination coverage for serotypes A and O of FMDV was evaluated using solid-phase competitive ELISA. The findings indicated that the FMD vaccinal strains utilized in the Abu Dhabi Emirate were antigenically matched with the field isolates. Moreover, the implemented FMD vaccination program with a booster dose elicited FMDV-specific antibody responses in sheep and goat herds with >80% coverage.
Abstract: Despite annual vaccination of livestock against Food and Mouth Disease (FMD) in the United Arab Emirates (UAE), outbreaks of the disease continue to be reported. Effective control of field outbreaks by vaccination requires that vaccines used are antigenically matched to circulating field FMD viruses. In this study, vaccine matching analysis was performed using two-dimensional virus neutralization test (VNT) for three field isolates belonging to the O/ME-SA/PanAsia-2/ANT-10 and O/SA-2018 lineages collected from different FMD outbreaks that occurred within the Abu Dhabi Emirate in 2021 affecting Arabian oryx (Oryx leucoryx), goat and sheep., In addition, post vaccination antibodies in sheep and goats were measured using solid-phase competitive ELISA (SPCE) for FMDV serotypes A and O at five months after a single vaccine dose and a further 28 days later after a second dose of FMD vaccine. Analysis of vaccine matching revealed that five out of the six vaccine strains tested, were antigenically matched to the UAE field isolates, with r1-values ranging between 0.32-0.48 and 0.32-0.75, respectively. These results suggest that the vaccine strains (O-3039 and O1 Manisa) included in the FMD vaccine used in the Abu Dhabi Emirate, are likely to provide protection against outbreaks caused by the circulating O/ME-SA/PanAsia-2/ANT-10 and O/ME-SA/SA-2018 lineages. All critical residues at site 1 and site 3 for VP1 were conserved in all isolates, although analysis of the VP1-encoding sequences revealed 14-16 amino acid substitutions compared to the sequence of the O1 Manisa vaccine strain. This study also reports on the results of post-vaccination monitoring where the immunization coverage against FMDV serotypes A and O was 47 % and 69 % five months post the first dose of FMD vaccination, and was increased to 81 and 88%, respectively, 28 days after the second dose of vaccination. These results reinforce the importance of using a second booster dose to reach maximize the impact of vaccination. In conclusion, the vaccine strains currently used in Abu Dhabi are antigenically matched to circulating field isolates from two serotype O clades (O/ME-SA/PanAsia-2/ANT-10 sublineage and O/SA-2018 lineages). The bi-annual vaccination schedule for FMD in Abu Dhabi Emirate has the potential to establish a sufficient herd immunity, especially when complemented by additional biosecurity measures for comprehensive FMD control in the Emirate of Abu Dhabi. These findings are pivotal for the successful implementation of the region's vaccination based FMD control policy showing that high vaccination coverage and wide-spread use of booster doses in the susceptible herds is required to achieve high level of FMDV-specific antibodies in the vaccinated animals
Line 93: Please update the epidemiology with the most recent publication.
Was updated to with recent reference provided as follows:
However, the region has also encountered introductions of O/ME-SA/Ind-2001 and A/ASIA/G-VII from Pool 2 (South Asia), SAT1 and SAT 2/XIV from Pool 4 (East Africa) .
The tow reference below were added to reference list
13- Aslam, M. and K.A. Alkheraije, The prevalence of foot-and-mouth disease in Asia. Frontiers in Veterinary Science, 2023. 10.
14 - Chepkwony, E.C., et al., Isolation and molecular characterization of Foot and Mouth Disease virus serotype O circulated in Kenya during the period 2013-2018.
Abbreviations: Please expand the full names for all abbreviations that are mentioned for the first time.
All abbreviations were expanded and highlighted
Tables and figures should be stand-alone; please describe all abbreviations in the footnote or ligands.
Tables and figure were included in the manuscript as its recommended by the journal format
Materials: Please provide the supplier, city, and Country of all materials used
Updated to be (Qiagen, Hilden, Germany)
Is serology enough for evaluating the efficacy of FMDV vaccines?
Serology may not be enough to evaluate the efficacy. This has been mentioned in the discussion part (While in vivo vaccination-challenge experiments, are considered the gold standard for FMD vaccine matching, they have limitations in terms of animal welfare, biosafety, and cost-effectiveness. In practice, FMD vaccine selection relies heavily on in vitro serological vaccine matching tests, such as virus neutralization tests (VNT) and liquid-phase blocking ELISA (LPBE)) and we avoided repeating the paragraph while discussing the post vaccination assessment results.
Additional minor comments: See attached pdf file
All comments were addressed and highlighted
Minor editing of English language required
Minor editing was carried out as highlighted

Reviewer 3 Report
Comments and Suggestions for Authors
The manuscript submitted by Eltahir and colleagues provides valuable information regarding the use and potential effectiveness of FMD vaccination in endemic regions. Here, the manuscript is well written and of sound methodology, and I recommend this manuscript for publication following minor revisions. Below, aspects of the manuscript requiring further clarity and revision are found:
-Lines 146-149: following swab and heart sample collection, how were the samples stored/maintained post-collection while sent to ADAFSA? For example, were the samples stored in a viral transport media or commercial preservation buffer?
-Lines 164-165: what threshold indicated a positive sample?
-Lines 201-202: regarding the homologous vaccine O strain, was this from six different commercial vaccines or is this one strain used in all six vaccines? It's somewhat unclear how currently written. Additionally, I recommend the author include the trade names of these six vaccines.
-Minor typographical and grammatical errors are found throughout the manuscript. For example, the comma after "that" on line 193, line 214 capitalization, "dyas" in Table 4.
Comments on the Quality of English LanguageMinor edits throughout the manuscript are required. Please find my comments and suggestions.
Author Response
Comments and Suggestions for Authors
The manuscript submitted by Eltahir and colleagues provides valuable information regarding the use and potential effectiveness of FMD vaccination in endemic regions. Here, the manuscript is well written and of sound methodology, and I recommend this manuscript for publication following minor revisions. Below, aspects of the manuscript requiring further clarity and revision are found:
Thanks for reviewing our manuscript and please find below our detailed response
-Lines 146-149: following swab and heart sample collection, how were the samples stored/maintained post-collection while sent to ADAFSA? For example, were the samples stored in a viral transport media or commercial preservation buffer?
The swab samples were collected in a viral transport media, while the tissue in a plain container. Both samples were immediately placed in ice box and transported directly to ADAFSA.
-Lines 164-165: what threshold indicated a positive sample?
The threshold indicated a positive sample is 39, this information was added to the manuscript.
-Lines 201-202: regarding the homologous vaccine O strain, was this from six different commercial vaccines or is this one strain used in all six vaccines? It's somewhat unclear how currently written. Additionally, I recommend the author include the trade names of these six vaccines.
Was corrected to: This serum is collected 21 days post-vaccination, and subsequently tested against both the homologous (six different serotype O commercial vaccinal strains produced by three different companies).
We regret providing the trade names of the vaccines. This is to avoid claims that could be raised by the production companies. Indeed, companies names providing the tested FMD serotypes (O) at the FMD -WRLB are presented in table 3
-Minor typographical and grammatical errors are found throughout the manuscript. For example, the comma after "that" on line 193, line 214 capitalization, "dyas" in Table 4.
The manuscript is revised, and all minors’ typographical errors were corrected and highlight.
Comments on the Quality of English Language, Minor edits throughout the manuscript are required. Please find my comments and suggestions.
The manuscript is revised, and all minors’ typographical errors were corrected and highlight
